# Synthesis of Small Peptide Nanogels Using Radiation Crosslinking as a Platform for Nano-Imaging Agents for Pancreatic Cancer Diagnosis

**DOI:** 10.3390/pharmaceutics14112400

**Published:** 2022-11-07

**Authors:** Atsushi Kimura, Tadashi Arai, Miho Ueno, Kotaro Oyama, Hao Yu, Shinichi Yamashita, Yudai Otome, Mitsumasa Taguchi

**Affiliations:** 1Takasaki Advanced Radiation Research Institute (TARRI), National Institutes for Quantum Science and Technology (QST), 1233 Watanuki-Machi, Takasaki 370-1207, Gunma, Japan; 2Graduate School of Science and Technology, Gunma University, 1-5-1 Tenjintyo, Kiryu 376-8515, Gunma, Japan; 3Nuclear Professional School, School of Engineering, The University of Tokyo, 2-22 Shirakata-Shirane, Tokai-mura, Naka-gun 319-1188, Ibaraki, Japan

**Keywords:** peptide, nanogel, radiation crosslinking, γ-rays, pancreatic cancer diagnosis, nanotechnology

## Abstract

Nanoparticle-based drug delivery systems (DDS) have been developed as effective diagnostic and low-dose imaging agents. Nano-imaging agents with particles greater than 100 nm are difficult to accumulate in pancreatic cancer cells, making high-intensity imaging of pancreatic cancer challenging. Peptides composed of histidine and glycine were designed and synthesized. Additionally, aqueous peptide solutions were irradiated with γ-rays to produce peptide nanogels with an average size of 25–53 nm. The mechanisms underlying radiation-mediated peptide crosslinking were investigated by simulating peptide particle formation based on rate constants. The rate constants for reactions between peptides and reactive species produced by water radiolysis were measured using pulse radiolysis. HGGGHGGGH (H9, H—histidine; G—glycine) particles exhibited a smaller size, as well as high formation yield, stability, and biodegradability. These particles were labeled with fluorescent dye to change their negative surface potential and enhance their accumulation in pancreatic cancer cells. Fluorescent-labeled H9 particles accumulated in PANC1 human pancreatic cancer cells, demonstrating that these particles are effective nano-imaging agents for intractable cancers.

## 1. Introduction

Imaging approaches such as magnetic resonance imaging (MRI) and positron emission tomography (PET) are used to detect early-stage disease and acquire detailed images in various medical conditions. Imaging agents must accumulate at specific locations, which usually requires drug delivery systems (DDS) and may produce side effects on patients [1]. Tumor tissues have significantly higher vascular permeability than normal tissues, allowing macromolecules and microparticles to easily escape from the blood vessels. These characteristics are known as the EPR (enhanced permeation and retention) effect, which is an important factor in the passive targeting of cancer cells [2]. In addition, smaller polymer particles are known to exhibit significant tumor targeting for specific uptakes [3]. Functional nanoparticles, therefore, can control drug release rate, target directionality, and drug absorbability, thereby enhancing drug efficiency and reducing side effects. In recent years, nanoparticle-based contrast agents that accumulate in cancer cells (passive targeting) have been developed for use in biological and medical imaging methods, including common fluorescence imaging, MRI, computed tomography (CT), ultrasonography (US), PET, and single-photon emission computed tomography (SPECT) [2,3,4,5,6,7,8,9]. Nanomaterials such as liposomes, albumin, metal particles, chemically crosslinked polymer particles, micelles, dendrimers, and emulsions have been developed as platforms for nano-imaging [10,11,12,13,14,15]. However, only chemically stable biomaterials, such as liposomes or albumin, have reached practical applicability as nano-imaging agents. Although many nano-imaging agents with excellent drug delivery properties have been developed at the laboratory level, these agents have not reached clinical research and practical applications owing to the bottleneck of the biocompatibility and chemical stability of their base materials. Micelles, emulsions, and dendrimers are biocompatible materials with easily controlled particle sizes; however, advanced molecular design and polymer matrix synthesis are required to enhance chemical stability. Chemically crosslinked polymer and metal particles are chemically stable and have easily-controlled particle sizes, but their biocompatibility is low. Specifically, it is difficult to target pancreatic cancer cells using nano-imaging agents [16,17,18]. Developing functional nanoparticle-based drugs with good biocompatibility, chemical stability, and particle size control is required to enhance pancreatic cancer diagnosis and treatment.

The radiation modification technique involving soluble proteins/peptides has been used to produce functional bio-devices from the nano- to macro-scale, without using cytotoxic chemical agents [19,20]. Solid-state proteins/peptides are decomposed by the direct effects of ionizing radiation [21]. The direct effects of ionizing radiation on proteins in water are suppressed because water molecules absorb most of the ionizing radiation energy. Furthermore, hydroxyl (OH) radicals with high oxidizing effects are generated by water radiolysis and attack the protein to form radicals. Recombination reactions between protein radicals create covalent bonds, forming a gel with a three-dimensional network [22]. Radiation-crosslinked proteins have been investigated as biomaterials for cell culture and DDS applications. Thus, radiation crosslinked protein/peptide hydrogels containing a large amount of water inside their three-dimensional structure are highly flexible materials, with both liquid and solid properties, and they are promising for applications in medicine, pharmacy, and biology.

We previously investigated changes in the chemical structure of gelatin in water irradiated with ionizing radiation using amino acid composition analysis [21]. Phenylalanine, tyrosine, and histidine residues in gelatin were key for radiation crosslinking, while other amino acid residues were barely reduced by ionizing radiation. The radiation crosslinking efficiency was accurately evaluated by qualitatively and quantitatively analyzing dityrosine, which occurs as the crosslinking point of the protein. Peptides composed of phenylalanine, tyrosine, and histidine were designed and synthesized to construct a nanoparticle platform providing imaging agents for cancer diagnosis. The synthesized peptides in water were irradiated to produce 100–500 nm peptide nanogels with good biodegradability, long-term chemical stability, and positive surface potential [23]. The surface charge of the nanogels is usually evaluated using the zeta potential. Positively charged nanoparticles (>10 mV) aggregate with serum proteins [24]. Therefore, the surface potential of the peptide nanoparticles was changed to a negative charge by binding fluorescent dye to their amino groups. The fluorescent-labeled nanogels were cultured with HeLa cells to confirm that the particles were taken up by the cells. Thus, peptide nanogels have potential applications as new tumor-targeted diagnostic agents and DDS carriers.

For cancer diagnosis and treatment, it is necessary to set peptide nanogel diameters to sizes that permeate the pancreatic stroma. Pancreatic cancer exhibits very high malignancy. Although the pancreatic stroma-permeable size depends on the individual patient and cancer progress, nanoparticles smaller than 100 nm can easily penetrate cancer cells [25]. In addition, nanogels need to have a negative charge to enhance intracellular accumulation because pancreatic cancer cell membranes have a positive charge [26,27,28]. This study aimed to produce radiation-crosslinked peptide nanogels smaller than 100 nm with a negative surface potential that can be used for pancreatic cancer diagnosis. Peptides consisting of histidine (involved in radiation crosslinking) and glycine (unrelated to radiation crosslinking), with different polymer chain lengths, were synthesized. Subsequently, aqueous peptide solutions were irradiated with γ-rays to produce nanogels with an average diameter of less than 100 nm. The radiation crosslinking-mediated mechanism of peptide nanogel formation in water was clarified using pulse radiolysis, molecular orbital calculations, and simulations. The stability, biodegradability, and accumulation of peptide nanogels in pancreatic cancer cells were then evaluated regarding their applications as nanogel-based drugs for pancreatic cancer diagnosis.

## 2. Materials and Methods

### 2.1. Reagents

The 2-Chlorotrityl chloride resin (reaction point density: 1.6 mmol g^−1^; Watanabe Chemical Industries, Hiroshima, Japan) was used as an insoluble resin carrier. To extend the peptide chain, *N*-Fmoc-*N*-trityl-L-histidine (Fmoc-His, >99%, Sigma-Aldrich, St. Louis, MA, USA), and 9-fluorenylmethyloxycarbonyl-glycine (Fmoc-Gly, AAPPTEC) were used. Selected as condensing agents were 1-[Bis(dimethylamino)methyliumyl]-1H-1,2,3-triazolo [4,5-b] pyridine-3-oxide hexafluorophosphate (HATU, AAPPTEC) and diisopropylethylamine (DIEA, Fujifilm Wako Pure Chemicals, Osaka, Japan). *N*,*N*-dimethylformamide (DMF), dichloromethane (DCM), methanol (MeOH), piperidine, trifluoroacetic acid, triisopropylsilane (TIS), and diethyl ether (Et_2_O) were purchased from Fujifilm Wako Pure Chemicals and were used for washing, cutting, and dissolving the Fmoc reagent. Ninhydrin and ethanol (both from Fujifilm Wako Pure Chemicals) were used for the ninhydrin color tests.

The designed and synthesized peptides are presented in Table 1. The procedure for peptide synthesis using a solid-phase column was reported in our previous work [23]. The synthesized peptides were dissolved in water (100 µL) and purified using a 0.22 µm membrane filter (Milex GS, Merck Millipore, Burlington, MA, USA), before and after irradiation, and analyzed using high-performance liquid chromatography connected to a fluorescence detector (2475, Waters, Milford, MA, USA) and a mass spectrometer (LCMS-2020, Shimadzu, Kyoto, Japan). Next, 0.1 wt.% formic acid and acetonitrile were eluted into the column (ODS-DE613, Shodex, Yokohama, Japan) with a gradient mixture ratio of formic acid/acetonitrile = 95/5 to 5/95 and a flow rate of 1.0 mL min^−1^ for HPLC-MS analysis.

### 2.2. Preparation and Irradiation of the Aqueous Synthetic Peptide Solution

Synthesized peptides were dissolved at 0.1 wt.% in Millipore Milli-Q water. The sample solutions were irradiated under aerated conditions at 25 °C, with γ-ray doses from 5–15 kGy (Gy = J kg^−1^) and dose rates from 0.5–10 kGy h^−1^ using a 60 Co γ-ray source at the Takasaki Advanced Radiation Research Institute, National Institutes for Quantum Science and Technology (QST). Dosimetric measurements of the absorbed dose and dose rate were carried out with an alanine-based dosimeter, as previously described [23].

After γ-ray irradiation, the aqueous peptide solution was filtered using ultrafiltration (Amicon Ultra, Merck, Kenilworth, NJ, USA) to separate particles from the liquid component. The particles were dried in a vacuum oven (VT220P, Kusumoto Chemicals, Tokyo, Japan) at 40 °C for 24 h and weighed using an electronic balance (BM-20, A&D Company, Tokyo, Japan) to estimate the peptide particle yield.

### 2.3. Pulse Radiolysis Study

Pulse radiolysis experiments were performed using an S-band linear accelerator (LINAC) at the Nuclear Professional School, University of Tokyo. A beam of 35 MeV electrons with a 10 ns pulse width at room temperature was used for radiolysis. The absorbed dose was estimated to be 4–4.9 Gy/pulse using the extinction coefficient of (SCN)_2_^−^ (ε = 7100 L mol^−1^ cm^−1^ at 472 nm) in an aqueous KSCN solution at 10 mmol L^−1^ under saturated nitrous oxide (N_2_O) conditions. The sample solutions were bubbled with N_2_O gas for at least 15 min.

### 2.4. Molecular Orbital Calculations and Kinetics Simulations

The stable structure of each peptide was determined using Spartan’18 software (Wavefunction) according to the density functional theory. The octanol/water partition coefficient (log *P*) of the peptides was estimated based on the determined stable structures. Peptide decomposition and crosslinking in water were simulated by modeling kinetics using FACSIMILE software (MCPA Software, Faringdon, UK).

### 2.5. Dynamic Light Scattering (DLS) and Electrophoretic Light Scattering (ELS) Measurements

The particle size and zeta potential of the peptide solution in water/phosphate-buffered saline solution (PBS; pH = 7.4; Gibco, Carlsbad, CA, USA), before and after γ-ray irradiation, were measured using DLS and ELS with a Zetasizer instrument (Malvern Panalytical, Worcestershire, UK) at room temperature (25 ± 1 °C). All data were analyzed as number-weighted and intensity-weighted values of the hydrodynamic diameter. Solvent viscosities of the sample solutions were measured with a viscometer (VM-10A, CBC Co., Ltd., Osaka, Japan) for the DLS and ELS measurements. The refractive index (RI) for the peptides was assumed to be that of a protein (RI = 1.45), and the refractive index of water was set to be RI = 1.33 [29]. The experimental data of all samples for DLS and ELS measurements were averaged for 3 experimental datasets. Accumulation time per experimental data was set at least 11 times. Nanogel stability in PBS was observed at 37 °C for 6 days.

### 2.6. Biodegradability

The peptide nanogel biodegradability in enzyme-containing buffer solution was investigated. The peptide nanogel solution (10 mL) was added to 10 mL 0.02 mol L^−1^ sodium hydrogen carbonate (guaranteed reagent grade, Fujifilm Wako Pure Chemical), 1 mmol L^−1^ calcium chloride (90% purity, Fuji-film Wako Pure Chemical), and 0.01 wt.% protease (from *Aspergillus oryzae*, Tokyo Chemical Industry, Tokyo, Japan). The mixed solutions were incubated at 37 °C for 6 days. Then, the absorbance of the treated solutions was measured using a spectrophotometer (U-3310 spectrophotometer, Hitachi, Tokyo, Japan). The degradation rate was determined from the absorbance at 273 nm using Equation (1):Degradation rate = Abs_1_/Abs_0_ × 100(1)
where Abs_1_ and Abs_0_ are the absorbance values of the filtrate and residue, respectively.

### 2.7. Cellular Uptake Tests

Fluorescent staining of the peptide nanogel solution was carried out using a Hi-Lyte Fluor 555 Labeling Kit-NH_2_ (LK14; Dojindo Laboratories, Kumamoto, Japan). The labeled peptide solution was poured into an ultrafiltration tube (3 kDa; Millipore) and washed three times with PBS by centrifugation (CF15RN, HIMAC, Berlin, Germany) at 7500 rpm. The fluorescent spectra of the labeled peptide nanogel solutions were obtained using a fluorescent spectrophotometer (RF-5300, Shimadzu).

HeLa human cervical cancer cells (RCB0007, Bioresource Center, RIKEN, Ibaraki, Japan) were cultured in Eagle’s minimum essential medium (Sigma-Aldrich, M5650) with 10 vol% fetal bovine serum (SH30910.03, Hyclone, Logan, UT, USA), 292 mg L^−1^ L-glutamine, 100,000 U L^−1^ penicillin, and 100 mg L^−1^ streptomycin (10378016, Thermo Fisher Scientific, Waltham, MA, USA). PANC-1 human pancreatic cancer cells (RCB2095, Bioresource Center) were cultured in RPMI 1640 medium (11875093, Thermo Fisher Scientific) supplemented with 10 vol% fetal bovine serum (12483020, Thermo Fisher Scientific), 100,000 U L^−1^ penicillin, and 100 mg L^−1^ streptomycin (15140122, Thermo Fisher Scientific). Cells were seeded in glass-bottom dishes (3911-035, AGC Techno Glass, Shizuoka, Japan) and incubated at 37 °C in 5% carbon dioxide for 1–2 days. After removing the medium, fluorescent-labeled peptide nanogels in PBS (75 μL) were mixed with Opti-MEM cell culture medium (31985070, Thermo Fisher Scientific; 75 μL) and placed in the dish. After incubation at 37 °C in 5% carbon dioxide for 2 or 22 h, the medium was changed to Opti-MEM medium containing 5 mg L^−1^ CellMask Green plasma membrane stain (C37608, Thermo Fisher Scientific). The cells were then incubated at 37 °C in 5% carbon dioxide for 5 min. After washing twice with PBS, the cells were incubated in 4 wt.% paraformaldehyde in PBS (163-20145, Fujifilm Wako Pure Chemicals) at room temperature for 10 min. Fixed cells were washed twice with PBS, incubated in PBS, and imaged using an upright microscope (BX51WI, Olympus, Tokyo, Japan) with a confocal disk scan unit (DSU, Olympus), an X60 water immersion lens (LUMPLFLN60XW, Olympus), and a COMS camera (ORCA-Flash4.0 V3, Hamamatsu Photonics, Shizuoka, Japan).

### 2.8. Statistical Analysis

The diameter and the zeta potential of all samples used for DLS and ELS measurements were averaged over at least 3 experimental datasets. The formation yield, stability, and biodegradability of the nanogels were calculated by averaging at least 3 experimental datasets. The absorption spectra of mixed aqueous solutions of KSCN, with the peptide measured using pulsed electron irradiation, were averaged across at least 20 experimental datasets.

## 3. Results

### 3.1. Radiation Crosslinking of Peptides with Different Chain Lengths in Water

The amino acids involved in radiation crosslinking in proteins are phenylalanine (Phe, F), tyrosine (Tyr, Y), and histidine (His, H) [21]. Glycine (Gly, G), the major amino acid residue, is not related to radiation crosslinking and suppresses steric hindrance in the peptide chain. Five peptides consisting of amino acids involved in radiation crosslinking and Gly were synthesized, and their aqueous solutions were irradiated with γ-rays to produce nanogels from 100 to 500 nm, as described previously [23]. Peptide nanogels containing Phe or Tyr, which have higher rate constants for OH radicals than His, yield larger particle sizes under the same irradiation conditions. Radiation-crosslinked nanogels containing Phe or Tyr showed low stability in PBS because of their hydrophobicity. When exposed to ionizing radiation, oligopeptides must have three or more histidine residues to form nanogels with a three-dimensional network structure [23]. Because peptides composed of four or more histidine amino acid residues enhance radiation crosslinking reactivity, the size of the peptide nanogel increases significantly. To produce peptide nanogels with a diameter of less than 100 nm, the peptides were composed of three histidines and a varying number of glycines, with lower radiation crosslinking reactivity. Therefore, HGGGHGGGH (H9) and HGGGGGHGGGGGH (H13), which contained additional Gly residues and were based on the stable peptide HGHGH (H5), were synthesized, as shown in Table 1. These aqueous peptide solutions were irradiated to produce nanogels for applications in pancreatic cancer diagnosis.

Aqueous H5, H9, and H13 solutions at 0.1 wt.% were irradiated with γ-rays under aerated conditions to produce peptide nanogels with an average size of 53, 49, and 30 nm by the number-weighted distribution analysis, respectively (Figure 1). The number-weighted distributions of H5, H9, and H13 (Figure 1a) show a smaller particle size than the intensity-weighted distribution (Figure 1b). The average polydispersity index (PDI) of these nanogels was 0.19, 0.22, and 0.23, respectively. These peptides are completely dissolved in water. Their aqueous solutions were transparent, and peak light scattering intensity was not detected before γ-ray irradiation. Therefore, the peak light scattering intensities measured by DLS indicate that peptide nanogels were produced from aqueous peptide solutions after γ-ray irradiation (Figure 1).

The dose-dependent particle size was next investigated (Figure 2). The H5, H9, and H13 particle diameters decreased slightly under high radiation doses. The average particle size of H13, which was analyzed by the number-weighted distribution, was significantly small at 15 kGy (Figure 2a), indicating that H13 nanogels were decomposed by the ionizing radiation. The increased particle size in the low-dose regions is due to the radiation crosslinking-induced peptide addition to the peptide chains, while the decreased particle size in high-dose regions may be due to particle radiolysis.

To investigate the differences in nanoparticle size (H5 > H9 > H13) at 5 kGy, the molecular structure and hydrophilic/hydrophobic properties of each peptide were evaluated by molecular orbital calculations using Spartan’18 software. The density functional theory (DFT, B3LYP, 6-31G**), which is used to analyze the structure of biomolecules, such as proteins, was used for these calculations [30]. The amino acid structures (bond lengths and ϕ and ψ values) were determined from previous studies [31,32]. The calculations assumed that the peptides were dissolved in water and that there were no interactions between molecular chains. The stable α-helix structure of each peptide chain was determined. Then, the log *P* values were estimated (Table 2). The peptide hydrophilicity increased with increasing Gly content in the peptides.

Peptide nanogel formation via radiation crosslinking is dependent on peptide multimerization resulting from the dimerization of aromatic amino acid residues [21,23]. Furthermore, multimeric peptide aggregation increased with increasing hydrophobicity, and dense peptide nanogels were produced from hydrophobic peptides. The H13 nanogels exhibited longer peptide chains, higher hydrophilicity, and smaller particle size, indicating a lower number of H13 molecules in the nanogels (i.e., the particle density was low). Therefore, the mechanisms underlying H5 and H9 nanogel formation are discussed in this study.

The H5 nanogel yield increased with increasing irradiation, but decreased after 10 kGy irradiation (Figure 3). The H9 yield increased with increasing radiation doses, reaching a maximum of 23.5%. The differences between the H5 and H9 yields may be due to different radiation crosslinking mechanisms. Therefore, the kinetics of H5 and H9 formation were studied using pulse radiolysis to estimate the rate constant of the OH radicals. Additionally, the crosslinking mechanisms of these peptides were simulated and analyzed.

### 3.2. Aqueous Peptide Solution Kinetics after Ionizing Radiation

The mechanism of radiation crosslinking between peptides and gelatin in water was reported previously [21,22,23]. The radiation crosslinking of solid-state peptides is unlikely [21], and the direct effect of γ-rays on peptide crosslinking is small. For aqueous gelatin solutions, radiation-induced crosslinking is due to the dimerization of amino acid radicals, which are produced by reactions between aromatic amino acid residues, such as phenylalanine, and reactive species produced from water radiolysis [21,22]. The reactive species are hydroxyl (OH) radicals, hydrated electrons (e^−^_aq_), and hydrogen atoms (H) (Equation (2)).
H_2_O 
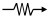
 OH + e^−^_aq_ + H (2)

The radiation yields (*G*-values) of these active species are 0.287, 0.27, and 0.055 μmol L^−1^ Gy^−1^ [33,34]. e^−^_aq_ and H react with dissolved oxygen to produce O_2_^−^ and HO_2_, which display lower reactivity than OH radicals. Therefore, the OH radical is the main reactive species contributing to the peptide crosslinking under aerated conditions [21,22,23].

The time-resolved spectra of mixed aqueous solutions of peptides and potassium thiocyanide (KSCN) were measured by pulsed electron radiolysis to evaluate OH radical reactivity. Nitrous oxide dissolved in water reacts with e^−^_aq_ and is converted into OH radicals (Equation (3)) [35,36].
e^−^_aq_ + N_2_O → ·OH + N_2_ + OH^−^(3)

Therefore, the radiation chemical yield of OH radicals under saturated nitrous oxide conditions was 0.567 mol L^−1^ Gy^−1^, which is suitable for observing reactions between peptides and OH radicals. The standard substance, KSCN, is completely ionized in water, while thiocyanate ions react with OH radicals to form dimer radicals, (SCN)_2_^−^·, as illustrated in Equations (4) and (5) [33].
SCN^−^ + OH → SCN· + OH^−^    *k*_SCN_ = 2.8 × 10^10^ mol^−1^ L s^−1^(4)
SCN^−^ + SCN· → (SCN)_2_^−^·    *k* = 7.0 × 10^9^ mol^−1^ L s^−1^(5)

(SCN)_2_^−^· has a characteristic optical absorption band at 472 nm (ε = 7100 mol^−1^ L cm^−1^). Thus, the reactivity of OH radicals can be analyzed from the time dependence of the absorption spectrum.

As demonstrated by pulse radiolysis, the absorption spectra for the mixed aqueous solutions of the peptide with KSCN at 500 nm were redshifted from 472 nm to prevent the overlap of absorption of (SCN)_2_^−^· with secondary products of the peptide [35]. The absorbance of 1 mmol L^−1^ KSCN at 500 nm increased immediately after pulsed electron irradiation and saturated after approximately 2 μs (Figure 4, solid line). The absorbance at 500 nm gradually decreased over time.

The absorbance of (SCN)_2_^−^· at 500 nm decreased with increasing H5 in the aqueous KSCN solution (Figure 4). These results indicate that reactions between SCN^−^ or H5 with OH radicals (Equations (4) and (6)) occur competitively.
HGHGH + OH →→ [HGHGH-OH](6)

The ratio of the absorbance of the mixed KSCN and H5 aqueous solution (Abs) to that of the KSCN aqueous solution (Abs_0_) was plotted as a function of the [H5] to [SCN^−^] ratio (Figure 5).

The rate constant of H5 with OH was calculated as 4.2 × 10^9^ mol^−1^ L s^−1^, based on Equation (7): (7)$$\frac{\mathrm{Abs}}{{\;\mathrm{Abs}}_{0}}=1+\frac{k_{H5}\;[H5]}{k_{\mathrm{SCN}}{\;[\mathrm{SCN}}^{-}]}$$
where *k*_H5_ is the rate constant of the reaction of the H5 with OH radicals, and *k*_SCN_ is the rate constant of the reaction of SCN^−^ with the OH radicals (Equation (4)). The rate constant of H9 with OH radicals was estimated as 2.7 × 10^9^ mol^−1^ L s^−1^ (Table 3). The rate constants for H5 and H9 were slightly lower than those for the single amino acid histidine (5.0 × 10^9^ mol^−1^ L s^−1^). The rate constant for H9 was smaller than that of H5 because H9 contains more glycine residues, which are less reactive with OH. The rate constant of H5 with OH radicals was estimated at *k* = 2.6 × 10^9^ mol^−1^ L s^−1^ via a competitive reaction method using HPLC, as described in a previous study [23]. A slightly larger value was obtained by the pulse radiolysis method used in this study. For the competition reaction method using HPLC, the ratio of the decrease in peptides to phenylalanine standard was evaluated after γ-ray irradiation to estimate the rate constant of the peptide with OH radicals. The rate constant estimated by the competition reaction method was considered for side reactions, such as reactions between the secondary product and the OH radicals. Therefore, the radiation crosslinking of peptides in water was simulated using the rate constant estimated by the pulsed radiolysis method.

### 3.3. Simulation of Radiation Crosslinking of Peptides in Water by Ionizing Radiation

The primary reactions of the active species produced from water radiolysis were previously reported in detail (Equation (2)) [37,38]. Aromatic amino acid residues in the peptide react with OH radicals to produce OH-adduct peptide radicals (Equation (8)). These radicals dimerize to form crosslinking products (Equation (9)).
Peptide + OH → [Peptide-OH]· (8)
[Peptide-OH]· + [Peptide-OH]· → Peptide-Peptide + 2H_2_O (9)

The rate constants of H5 and H9 with OH radicals used for the simulation are listed in Table 3. The rate constants for dimerization between peptide or protein radicals were reported to be between 10^3^ to 10^5^ mol^−1^ L s^−1^ [39,40,41]. Therefore, the rate constant for peptide radical dimerization was set at 10^4^ mol^−1^ L s^−1^ for the simulation.

**Table 3 pharmaceutics-14-02400-t003:** Rate constants of peptides with OH radicals in water as evaluated by the KSCN method.

Peptide	Rate Constant with Hydroxyl Radicals (mol^−1^ L s^−1^)
H	5.0 × 10^9^ *
H5	4.2 × 10^9^
H9	2.7 × 10^9^
G	1.7 × 10^7^ *

* from Masuda et al. 1973 [38].

The peptide dimer shown in Equation (9) reacts with the OH radicals to form dimer radicals, which repeatedly react with peptide monomers to form polypeptides (Equation (10)).
Peptide-Peptide + OH → [Peptide-Peptide-OH]· + peptide →→→ Polypeptide(10)

However, polypeptides are degraded by OH radicals to form low-molecular-weight polypeptides (Equation (11)).
Polypeptide + OH → low-molecular-weight polypeptide(11)

The rate constant for the reaction of OH radicals with dimers of aromatic compounds was reported to be 10^9^–10^10^ mol^−1^ L s^−1^ [42,43,44], and the rate constant for polypeptide degradation by OH radicals was assumed to be 10^9^ mol^−1^ L s^−1^.

The concentration of dissolved oxygen in water under aerated conditions was calculated to be approximately 250 μmol L^−1^. The dissolved oxygen reacts with e^−^_aq_ and H to produce O_2_^−^ and HO_2_, as shown below (Equations (12) and (13)).
H + O_2_ → HO_2_(12)
e^−^_aq_ + O_2_ → O_2_^−^(13)

The reaction of OH radicals with aromatic compounds is accelerated by dissolved oxygen [45]. Peptide radicals were formed by the reaction of an aromatic peptide with OH radicals, as shown in Equation (8). These radicals react with dissolved oxygen to produce an OH-substituted peptide (Equation (14)).
[Peptide-OH]· + O_2_ → Peptide-OH + HO_2_
(14)

The OH-substituted peptide was assumed to react with OH radicals to form an open-ring (Equation (15)). The reaction rate constant was set at 10^9^ mol^−1^ L s^−1^, based on the information contained in previous literature [45,46,47].
Peptide-OH + OH → Open-ring products(15)

The change in the crosslinking density of H5 or H9 as a function of absorbed γ-rays was simulated based on Equations (8)–(15), as shown in Figure 6.

The simulated crosslinking densities of H5 and H9 increased with the absorbed dose and then were stable up to 15 kGy. Histidine-containing peptide nanogels were formed by the reactions summarized in Figure 1.

The OH radicals generated by water radiolysis reacted with the imidazole ring in the histidine-containing peptides to form peptide radicals. These molecules could dimerize (crosslinking) or undergo OH substitution reactions (decomposition). The peptide nanogel aggregation produced by irradiation with γ-rays was suppressed because H5 and H9 are hydrophilic (Table 1). Therefore, peptide dimers repeatedly reacted with peptide monomers to form peptide nanogels. OH-substituted products were further oxidized by OH radicals to form open-ring products. Additionally, some peptide nanogels were degraded by their reaction with OH radicals.

### 3.4. Peptide Nanogel Accumulation in Pancreatic Cancer Cells

The number-weighted distribution of only PBS exhibited intense peaks in the range from 0.5 to 2 nm, derived from impurities/contaminants. The number-weighted distribution of peptide nanogels was not obtained in PBS. Figure 1 and Figure 2 show that there is a difference in the absolute values of particle sizes between the number-weighted distribution and the intensity-weighted distribution, and a trend can be noted. Therefore, in this work, the stability of the peptide nanogels in PBS was evaluated by the intensity-weighted distribution. The peptide nanogels should measure less than 100 nm and should have a negative surface potential to facilitate their accumulation in pancreatic cancer cells. The H9 nanogels were extremely stable in PBS for 6 days. This experiment simulated in vivo conditions. However, the H5 and H13 particle sizes changed significantly under similar conditions (Figure 7). The peptide nanogel stability occurred due to its surface potential and resistance to oxidation. Particles with a positive or negative surface charge were more likely to disperse because of electrostatic repulsion, while particles without a surface charge were more likely to aggregate. H5 was less hydrophilic than H9, and the size of H5 nanogels was increased owing to aggregation. H13, which was more hydrophilic than H5 and H9, had a lower particle density and was likely oxidized by dissolved oxygen because peptides containing tyrosine are oxidized by dissolved oxygen in water and PBS [23].

The biodegradability of H5 nanogels was previously reported [23]. The degradation rates of the H9 and H13 nanogels for protease reactions after 6 days were 91 and 93%, respectively. Additionally, the half-lives of H9 and H13 in PBS were calculated to be 1.5 and 1.7 days, respectively. Proteases cleave covalent bonds in the main peptide chain. Since the crosslinking structure between the peptide side chains was formed by ionizing radiation, the structure of the main chain of the peptide did not change. Therefore, radiation-crosslinked peptide nanogels retain their biodegradability.

H9 nanogels, which have a smaller particle size, high formation yield, high chemical stability, and good biodegradability, accumulated in pancreatic cancer cells. The zeta potential of the H9 nanogels changed from +23.7 to −25.0 mV after fluorescence labeling, likely due to decreased positively-charged amino groups in the nanogels. Thus, fluorescently labeled H9 nanogels show enhanced accumulation in pancreatic cancer cells.

Fluorescent H9 nanogels accumulated in the cytoplasm of HeLa cells (Figure 8a) and human pancreatic PANC1 cancer cells (Figure 8b). No notable cellular morphological changes were observed, indicating that the nanogels exhibit low toxicity. The observed bright spots were larger than the size of the individual nanogels, suggesting that the nanogels entered the cells via endocytosis and accumulated in the intracellular vesicles, such as endosomes and lysosomes. The negative surface potential of the H9 nanogels enhances adhesion to pancreatic cancer cells, while a particle size less than 100 nm allows for accumulation inside the pancreatic cancer cells via endocytosis. The nanogels accumulated in only 2 h, which is faster than their half-life when degraded by a protease. These features may be useful to improve the efficacy of MRI or PET imaging of intractable cancers.

## 4. Conclusions

H5, H9, and H13 nanogels, which are composed of peptides containing histidine (involved in radiation crosslinking) and glycine, were designed and synthesized. Aqueous peptide solutions were irradiated by γ-rays at 5 kGy to produce nanogels with average diameters of 53, 49, and 30 nm, respectively. The sizes of the H5, H9, and H13 nanogels decreased with increasing γ-ray absorption. Moreover, nanogel decomposition appeared to occur. The stable structure of each peptide was determined using molecular orbital calculations to elucidate differences in particle sizes. The log *P* values for these peptides were calculated. The hydrophilicity of the peptide increased with increasing glycine content. Increased peptide hydrophobicity enhanced aggregation to form high-density nanogels. These results indicate that hydrophilic H13 nanogels with the lowest log *P* value exhibit low density, as well as chemical stability.

The rate constants of H5 and H9 (hydrophobic and high-density) peptides with OH radicals were determined using pulse radiolysis to investigate the underlying crosslinking mechanisms. The crosslinking densities of these peptides in water as a function of radiation dose were simulated using the predicted rate constants. OH radicals produced by water radiolysis react with imidazole rings in histidine-containing peptides to form peptide radicals, which undergo dimerization (crosslinking) and OH-substitution reactions (decomposition). The OH-substituted products were further decomposed by OH radicals to form open-ring compounds.

Peptide nanogels less than 100 nm in diameter and with a negative surface potential accumulated in pancreatic cancer cells. H9 nanogels were extremely stable in PBS, whereas H5 and H13 nanogels were unstable. The H9 biodegradability was 91% after 6 days. Thus, H9 nanogels are suitable nanosensors for diagnosing pancreatic cancer because of their small particle size, high formation yield, stability, and biodegradability.

The surface potential of fluorescent-labeled H9 nanogels was measured at −25.0 mV. Importantly, these modified nanogels accumulated in pancreatic cancer cells. Our results show that we were successful in developing peptide nanogels that can be used as a platform for nano-imaging agents for pancreatic cancer diagnosis.

In this work, the accumulation of the peptide nanoparticles to only one type of pancreatic cell has been carried out. However, the pancreatic cancer cells are surrounded by other structures, such as cancer-associated fibroblasts, in real-case situations. Further accumulation experiments using the nanogels will be conducted on multicellular spheroids and organoids [48,49,50]. Moreover, in vivo experiments using pancreatic tumor-bearing mice will be carried out to investigate the differences in the accumulation effect between healthy and pancreatic cancer organs. To enhance selective cell adhesion, amino acid motifs with cell adhesion properties, such as the RGD motif, should be incorporated into the H9 nanogels. The concentration effect and the quantitative result of the cellular internalization of the nanogels is required for practical usage, particularly in the development of diagnostic tools. Instead of fluorescent dyes, we plan to load gadolinium or radioisotopes into the H9 nanogels and observe their accumulation in vivo by MRI or PET. We believe these modified peptide nanogels will improve the nano-imaging of intractable cancers.

## Data Availability

Not applicable.

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
