# Peer review of "Synthesis of Small Peptide Nanogels Using Radiation Crosslinking as a Platform for Nano-Imaging Agents for Pancreatic Cancer Diagnosis"

_pharmaceutics, 2022, doi:10.3390/pharmaceutics14112400_

Round 1
Reviewer 1 Report
The manuscript present very interesting data. However the authors have used only one type of cells for the experiment. In real case situation the pancreatic cancer cells are surrounded by other structures such as cancer associated fibroblasts. What would the authors expect in such cases? Have they carried out experiments with a co-culture of fibroblasts and pancreatic cancer cells?
Have they conducted the in vitro experiments with healthy cells as control?
These in vitro experiments would add value to the manuscript.
Author Response
Thank you for your comments regarding our manuscript, pharmaceutics-1967346, entitled “Synthesis of small peptide nanogels using radiation crosslinking as a platform for nano-imaging agents for pancreatic cancer diagnosis”. We wish to express our appreciation to the reviewer for the insightful and constructive comments, which helped us revise the manuscript.

Reviewer 2 Report
The manuscript reports on the design of nanogel particles from small peptides by gamma-radiations. The experiments look OK; however, the data analysis was performed on a very low level. I can’t recommend the manuscript for publication.
My concerns:
- DLS data is of low level. Use number-weighted values of hydrodynamic diameter instead of intensity-weighted to draw any conclusions on the difference between H5, H9, and H13. I bet, H5 will have the smallest size in number-weighted distribution. Data for Figs. 2 and 7 also reanalyzed in number-weighted values.
- Authors use a wrong symbol for zeta potential. They use ksi symbol but not zeta on
- The details of DLS and ELS experiments should be provided; number of runs, accumulation time, solvent viscosity, filtration if any… Correlation functions should be provided.
- The calculation of crosslinking density is completely wrong. To start with, eq. 16 is not applicable for branched molecules nanogels. It’s valid for linear macromolecules in theta solvent only. There are limited number of polymer-solvent systems that can be adequately described by e freely joined chain model. For the rest of other systems you should take care of valent bond angles, excluded volume interactions etc. Secondly, b in eq. 16 is NOT a monomer unit length but Kuhn segment length that usually higher than a monomer unit length. The difference is usually 1.5-1.6 for flexible polymer chains and could be up to 200-300 for rigid ones. Finally, you can’t use hydrodynamic diameter value obtained from DLS experiments as R in eq. 16!!! It’s completely wrong.
- References 19-27 are self-citations.
Author Response

(The authors gave the same response as above.)

Round 2
Reviewer 1 Report
Since the authors did not carry any new experiments with coculture. It is suggested that the addition to the manuscript be listed as a limitation of the present study and how the authors plan to address this.
Author Response
Thank you for your comments regarding our manuscript. The manuscript has been rechecked and the necessary changes have been made in accordance with the suggestions.

Reviewer 2 Report
the manuscript can be published as is
Author Response
Thank you for your comments regarding our manuscript.